# Magnetic Interconnects Based on Composite Multiferroics

**DOI:** 10.3390/mi13111991

**Published:** 2022-11-17

**Authors:** Alexander Khitun

**Affiliations:** Electrical Engineering Department, University of California Riverside, Riverside, CA 92521, USA; akhitun@engr.ucr.edu

**Keywords:** synthetic multiferroic, interconnects, magnetic logic devices

## Abstract

The development of magnetic logic devices dictates a need for a novel type of interconnect for magnetic signal transmission. Fast signal damping is one of the problems which drastically differs from conventional electric technology. Here, we describe a magnetic interconnect based on a composite multiferroic comprising piezoelectric and magnetostrictive materials. Internal signal amplification is the main reason for using multiferroic material, where a portion of energy can be transferred from electric to magnetic domains via stress-mediated coupling. The utilization of composite multiferroics consisting of piezoelectric and magnetostrictive materials offers flexibility for the separate adjustment of electric and magnetic characteristics. The structure of the proposed interconnect resembles a parallel plate capacitor filled with a piezoelectric, where one of the plates comprises a magnetoelastic material. An electric field applied across the plates of the capacitor produces stress, which, in turn, affects the magnetic properties of the magnetostrictive material. The charging of the capacitor from one edge results in the charge diffusion accompanied by the magnetization change in the magnetostrictive layer. This enables the amplitude of the magnetic signal to remain constant during the propagation. The operation of the proposed interconnects is illustrated by numerical modeling. The model is based on the Landau–Lifshitz–Gilbert equation with the electric field-dependent anisotropy term included. A variety of magnetic logic devices and architectures can benefit from the proposed interconnects, as they provide reliable and low-energy-consuming data transmission. According to the estimates, the group velocity of magnetic signals may be up to 10^5^ m/s with energy dissipation less than 10^−18^ J per bit per 100 nm. The physical limits and practical challenges of the proposed approach are also discussed.

## 1. Introduction

The development of novel computational devices is well stimulated by the technological challenges and physical limits of the current complimentary metal–oxide–semiconductor (CMOS) technology [1]. Magnetic logic circuits are among the most promising approaches offering a significant reduction in consumed power by utilizing the inherent non-volatility of magnetic elements. In magnetic logic circuitry, logics 0 and 1 are encoded into the magnetization state of a nano-magnet, which may be kept for a long time without any power consumption, while the external energy is required only to perform computation (i.e., switching between the magnetization states). Though magnetic memory became a widely used commercial product a long time ago, magnetic logic is largely in its infancy. The development of energetically efficient and reliable magnetic interconnects is one of the main challenges to be overcome. Similar to electronic transistor-based circuits, where one transistor drives the next stage transistors by electric signals, magnetic logic circuits require one magnet to drive the next stage magnets by sending magnetic signals. There is a variety of possible mechanisms for magnetic signal transmission between the input and the output magnets (i.e., by making an array of nano-magnets sequentially switched in a domino fashion [2], by sending a spin-polarized current [3], by sending a spin wave [4], or by moving a domain wall [5]). There is always a tradeoff between the speed, the energy per bit, and the reliability of magnetic signal transmission. It takes either a large amount of energy for error-prone signal transmission, or the error probability increases with the distance due to the thermal noise, defects, and signal dispersion. The lack of amplification is one of the key issues inherent to the above-mentioned approaches. In this work, we consider composite multiferroics for magnetic interconnects, which may provide magnetic signal amplification by transferring energy between the electric and magnetic domains.

Composite multiferroics (or two-phase multiferroics) comprise piezoelectric and magnetoelastic materials, where an electric field applied across the piezoelectric produces stress, which, in turn, affects the magnetization of the magnetoelastic material. Although the study of composite multiferroics started in the 1970s [6], they have been in the shadow of the single-phase multiferroics (i.e., BiFeO_3_ and its derivatives [7]) for a long time. Currently, there is a resurgence of interest in composite multiferroics due to the technological flexibility in the independent variation of piezoelectric or magnetostrictive layers. The most important advantage of composite multiferroics over the single-phase ones (e.g., BiFeO_3_) is the larger strength of the electro-magnetic coupling, which can significantly exceed the limits of their single-phase counterparts [8]. Magnetization rotation in two-phase multiferroics was observed as a function of the applied voltage in several experimental works [9,10]. For instance, a reversible and permanent magnetic anisotropy reorientation was reported in a magnetoelectric polycrystalline Ni thin film and (011)-oriented [Pb(Mg_1/3_Nb_2/3_)O_3_](1 − x)–[PbTiO_3_] x (PMN-PT) heterostructure [9]. The application of a 0.2 MV/m electric field induces 1200 ppm strain, which, in turn, affects the magnetization of Ni film. According to our preceding work on a similar sample [11], a 0.8 MV/m electric field produces a linear response with in-plain anisotropic strains of εx = 350 μm/m and εy = −1200 μm/m. It is also important to note that the changes in magnetization states are stable without the application of an electric field and can be reversibly switched by an electric field near a critical value (i.e., 0.6 MV/m for Ni/PMN-PT). An ultra-low energy consumption required for magnetization rotation is possible because of this relatively small electric field [12]. The idea of using a stress-mediated mechanism for nano-magnet switching is currently under extensive study [13,14]. The development of multiferroics provides a new approach to spin-wave control. For instance, strain reconfigurable spin-wave transport in the lateral system of magnonic stripes was achieved [15]. It was also observed that the properties of spin-wave propagation in magnonic crystal in contact with a piezoelectric layer can be controlled by an external electric field [16]. Recently, spin-wave propagation and interaction were demonstrated in the double-branched Mach–Zehnder interferometer scheme. The use of a piezoelectric plate connected to each branch of the interferometer leads to the tunable interference of the spin-wave signal at the output section [17]. Here, we propose to utilize multiferroics in magnetic interconnects and exploit the strain-mediated electro-magnetic coupling for magnetic signal amplification. The rest of the paper is organized as follows. In Section 2, we describe the material structure and the principle of operation of the composite multiferroic interconnects. The results of numerical modeling illustrating signal propagation are presented in Section 3. The Discussion and Conclusions are given in Section 4 and Section 5, respectively.

## 2. Material Structure and Principle of Operation

The schematics of the proposed interconnect on top of a silicon wafer are shown in Figure 1A. It consists of the bottom to the top of a conducting layer (e.g., Pt), a layer of piezoelectric material (e.g., PMN-PT), and a layer of magnetoelastic material (e.g., Ni). The whole structure represents a parallel plate capacitor filled with a piezoelectric, where one plane (the bottom) comprises a non-magnetic metal and the top plate comprises a magnetoelastic metal. The top layer is the medium for magnetic signal propagation between the nano-magnets to be placed on the top of the layer. For simplicity, we have shown just two nano-magnets, which are marked as A and B in Figure 1A. The nano-magnet market A is the input element to send a magnetic signal to the receiver nano-magnet B. The spins of the nano-magnets are coupled to the spins of the ferromagnetic magnetostrictive layer via the exchange interaction. The nano-magnets are assumed to be of a special shape to ensure the two thermally stable states of magnetization. Hereafter, we assume the magnetoelastic layer to be polarized along the *x*-axis, and the nano-magnets to have two states of magnetization along or opposite the *y*-axis. Each of the nano-magnets has an electric contact where a control voltage is applied. The bottom layer, comprising a nonmagnetic metal, serves as a common ground plate.

The principle of operation is the following. In order to send a signal from A to B, a control voltage V is applied to the nano-magnet A. The application of voltage starts the charge diffusion through the conducting plates. The equivalent circuit is shown in Figure 1B. The charge diffusion through the capacitor plates is well described by the RC model, where the resistance R and the capacitance C are defined by the geometric size and the material properties of the conducting plates and the piezoelectric layer. An electric field appears across the piezoelectric produces stress, which affects the anisotropy of the magnetostrictive material by rotating its easy axis. It is assumed that the application of voltage rotates the easy axis from the *x*-axis towards the *y*-axis. The change of the anisotropy field caused by the applied voltage affects the magnetization of the magnetoelastic layer. There are two possible trajectories for the magnetization to follow: along or opposite the *y*-axis. The particular trajectory is defined by the magnetization state of the sender nano-magnet A (i.e., the magnetization of the ferromagnetic layer copies the magnetization of the sender nano-magnet).

In Figure 1C, we present the results of numerical modeling, showing the snapshot of the distribution of the electric field E(x) and the magnetization component My(x) through the interconnect. The details of numerical modeling are presented in the next section. Here, we wish to illustrate the main idea of using composite multiferroics as a magnetic interconnect: magnetic signals (i.e., the local change of magnetization) can be sent through large distances without degradation, as the angle of magnetization rotation is controlled by the applied voltage. The direction of signal propagation (e.g., from A to B, or vice versa) is also controlled by the applied voltages. The charging of the capacitor eventually leads to the uniform electric field distribution among the plates and the static distribution of magnetization through the magnetoelastic layer. There are several possible ways to switch output nano-magnet B. For example, it can be preset in a metastable state prior to computation (e.g., magnetization along the *z*-axis), so the magnetic signal sent by A triggers the relaxation towards one of the thermally stable states along or opposite to the *y*-axis. There may also be possible scenarios where the receiver nano-magnet is connected to two or more nano-magnets, so the final state is defined by the interplay of several incoming signals (e.g., MAJ operation). In this work, we focus on the mechanism of signal transmission only, though the utilization of composite multiferroic interconnects may further evolve the design of magnetic logic circuits similar to the ones presented in [3,4,18].

## 3. Numerical Modeling

The model for signal propagation in the composite multiferroic combines electric and magnetic parts. The electric part aims to find the distribution of an electric field through the piezoelectric, and the magnetic part describes the change of magnetization in the magnetoelastic layer. The charge distribution is modeled via the following equation [19]:(1)$$R_{s}C_{s}\frac{d^{2}V\left(x,t\right)}{dx^{2}}=\frac{dV\left(x,t\right)}{dt}$$
where *R_s_* and *C_s_* are the resistance and capacitance per unit length, and *V*(*x*,*t*) is the voltage distribution over the distance. The simulations start with *V*(0,0) = *V*in, and *V*(*x*,0) = 0 everywhere else through the plates.

The process of magnetization rotation is modeled via the Landau–Lifshitz equation [20]:(2)$$\frac{d\vec{m}}{dt}=-\frac{\gamma }{1+\eta^{2}}\vec{m}\times [{\vec{H}}_{eff}+\eta \vec{m}\times {\vec{H}}_{eff}]$$
where $\vec{m}=\vec{M}/M_{s}$ is the unit magnetization vector, *M_s_* is the saturation magnetization, *γ* is the gyro-magnetic ratio, and *η* is the phenomenological Gilbert damping coefficient. The effective magnetic field ${\vec{H}}_{eff}$ is the sum of the following:(3)$${\vec{H}}_{eff}={\vec{H}}_{d}+{\vec{H}}_{ex}+{\vec{H}}_{a}+{\vec{H}}_{b}\;$$
where *H_d_* is the magnetostatic field, *H_ex_* is the exchange field, *H_a_* is the anisotropy field ${\vec{H}}_{a}=\left(2K/M_{s}\right)(\vec{m}\cdot \vec{c})\vec{c}$ (*K* is the uniaxial anisotropy constant, and $\vec{c}$ is the unit vector along the uniaxial direction), and *H_b_* is the external bias magnetic field. The two parts are connected via the voltage-dependent anisotropy term as follows:(4)$$c_{x}=\cos\left(\theta \right),\;c_{y}=\sin\left(\theta \right),\;c_{z}=0$$
$$\theta =\frac{\pi }{2}\left(\frac{V(x)}{V_{\pi }}\right)$$
where *V_π_* is the voltage resulting in a 90-degree easy axis rotation in the X-Y plane.

The introduction of the voltage-dependent anisotropy field (Equation (4)) significantly simplifies simulations, as it presumes an immediate anisotropy response on the applied electric field without considering the stress-mediated mechanism of the electro-magnetic coupling. Such a model can be taken as a first-order approximation. Nevertheless, this model is useful in capturing the general trends of signal propagation and can provide estimates of the maximum speed of signal propagation and energy losses. In our numerical simulations, we use the following material parameters: the dielectric constant ε of the piezoelectric is 2000; the electrical resistivity of the magnetoelastic material is 7.0 × 10^−8^ Ω·m, the gyro-magnetic ratio is *γ* = 2 × 107 rad/s, the saturation magnetization is *M_s_* = 10 kG/4π; 2 K/Ms = 100 Oe, external magnetic field *H_b_* = 100 Oe is along the *x*-axis, and the Gilbert damping coefficient is *η* = 0.1 for the magnetostrictive material. For simplicity, we also assumed the same resistance for the bottom and the top conducting plates. The strength of the electro-magnetic coupling (i.e., *V_π_*) is calculated based on the available experimental data for PMN-PT/Ni (i.e., 0.6 MV/m for 90-degree rotation [9]). More details on the simulation procedure can be found in [21].

The results of the numerical simulations shown in Figure 1C are obtained for the interconnect comprising 40 nm of piezoelectric and 4 nm of magnetoelastic materials. The two curves in Figure 1C depict the distribution of the electric field *E*(*x*) and the projection of magnetization My(x) along the interconnect after the voltage has been applied through the nano-magnet *A*. The curves are plotted in the normalized units *E*/*E*_0_ and *My*/*Ms*, where *E*_0_ = *V_π_*/*d*, where d is the thickness of the multiferroic layer (40 nm). The distribution of the electric field was found by solving Equation (1). Then, the anisotropy field was found via Equation (4), and, finally, magnetization change was simulated via Equations (2) and (3). The results in Figure 1C show a snapshot taken at 0.4 ns after the voltage has been applied. In these simulations, we assumed the nano-magnet A to be polarized along the *y*-axis, and the magnetization of the interconnect beyond the nano-magnet My(0) = 0.1*M_s_* due to the exchange coupling with the spins of the nano-magnet. The spins of the magnetoelastic material tend to rotate in the same direction as the spins of the sender nano-magnet A. Eventually, the Y-component of the magnetization of the interconnect saturates along the constant value, which is defined by the interplay of the anisotropy and the bias magnetic fields.

In Figure 2, we show the results of numerical modeling illustrating the dynamics of magnetization rotation in the interconnect. The curves in Figure 2 depict the evolution of local magnetization in the interconnect located 1.5 µm, 1.7 µm, and 2.0 µm away from the excitation point. The insets in Figure 2 show the initial state of magnetization of the sender nano-magnet A. In all cases, the magnetization trajectory in the interconnect repeats the initial magnetization of the nano-magnet A (e.g., the magnetization component My is positive if nano-magnet A is polarized along the *y*-axis, and the My is negative if nano-magnet A is polarized opposite to the *y*-axis). The absolute value of the final steady state is the same (about 0.5 Ms) for all six curves. These results illustrate the main idea of implementing electric field-driven multiferroic interconnects, allowing us to keep the amplitude of the magnetic signal constant regardless of the propagation distance.

## 4. Discussion

The ability to pump energy into the magnetic signal during its propagation is the most appealing property of the described interconnects. The pumping occurs via the magneto-electric coupling in the multiferroic, where some portion of the electric energy provided to the capacitor is transferred to the energy of the magnetic signal. The amplitude of the magnetic signal (i.e., the angle of magnetization rotation) is controlled by the applied voltage and saturates to a certain value as the electric field across the piezoelectric reaches its steady-state distribution. This property is critically important for logic circuit construction, allowing us to minimize the effect of structure imperfections and make logic circuits immune to thermal noise. It should be also noted that the absolute value of magnetization change in the interconnect may exceed the initial magnetization state of the sender nano-magnet. For instance, the Y component of the magnetization of the nano-magnet A may be 0.1 Ms, while the Y magnetization of the magnetic signal in the interconnect may saturate around 0.5 Ms, as illustrated by numerical modeling in the previous section. In other words, the proposed interconnects may serve as an amplifier for magnetic signals, similar to the multiferroic spin-wave amplifier described in [21]. Another important property of the proposed interconnect is the ability to control the direction of signal propagation by the applied voltage. Similar to the “All Spin Logic” approach [3], where the direction of magnetic signal is defined by the direction of spin-polarized current flow, the change of magnetization in the multiferroic interconnect follows the charge diffusion. This property resolves the problem of input–output isolation and provides an additional degree of freedom for logic circuit construction.

Energy dissipation in a two-phase magnetoelastic/piezoelectric multiferroic has been studied in [14,22,23]. According to the estimates, a single two-phase magnetoelastic/piezoelectric multiferroic single-domain shape-anisotropic nano-magnet can be switched, consuming as low as 45 kT for a delay of 100 ns at room temperature, where the main contribution to the dissipated energy comes from the losses during the charging/discharging (≈CV^2^) [23]. The capacitance of one-micrometer-long multiferroic interconnects comprising 40 nm of PZT and 4 nm of Ni with the width of 40 nm is about 15 fF, and the control voltage required for 90-degree anisotropy easy-axis change is 0.6 MV/m × 40 nm = 24 mV. Thus, assuming all the electric energy dissipated during signal propagation, one has 9 aJ per signal per 1 µm transmitted. It is important to note that, according to the theoretical estimates [23], the energy dissipation increases sub-linearly with the switching speed. For example, in order to increase the switching speed by a factor of 10, the dissipation needs to increase by a factor of 1.6.

The propagation of the magnetization signal involves several physical processes: charge diffusion, the mechanical response of the piezoelectric to the applied electric field, change of the anisotropy field caused by the stress, and magnetization relaxation. Thus, the total delay time *τ_t_* is the sum of the following:(5)$$\tau_{t}=\tau_{e}+\tau_{mech}+\tau_{mag}$$
where *τ_e_* is the time delay due to the charge diffusion *τ_e_* = RC, τmech is the delay time of the mechanical response *τ_mech_* ≈ *d*/*v_a_*, where *d* is the thickness of the piezoelectric layer, *v_a_* is the speed of sound in the piezoelectric, and *τ_mag_* is the time required for the spins of magnetostrictive material to follow the changing anisotropy field. In the theoretical model presented in the previous section, we introduced a direct coupling among the electric field and the anisotropy field (i.e., Equation (4)), presuming an immediate anisotropy field response on the applied electric field. The latter may be valid for the thin piezoelectric layers (e.g., taking d = 40 nm, *v_a_* = 1 × 10^3^ m/s, *τ_mech_* is about 40 ps). We also introduced a high damping coefficient *η*, which minimizes the magnetic relaxation time *τ_meag_* < 50 ps. In this approximation, the speed of signal propagation is mainly defined by the charge diffusion rate. The smaller RC, the faster the charge diffusion and the lower the energy losses for interconnect charging/discharging.

In Figure 3, we show the results of numerical modeling on the speed of signal propagation for different thicknesses of the piezoelectric layer. The four curves correspond to the signal propagation in the interconnects with different PMN-PT thicknesses (e.g., 20 nm, 40 nm, 80 nm, and 200 nm), respectively. The thickness of the nickel layer is 4 nm for all cases. We also plotted a reference line corresponding to the magnetostatic spin wave with a typical group velocity of 3.1 × 10^4^ m/s. According to these estimates, one may observe that the magnetic signal in the multiferroic interconnect may propagate faster than the spin wave at short distances (<500 nm) and slower than the spin wave at longer distances. The latter leads to an interesting question of whether or not it is possible to transmit magnetic signals faster than the spin wave in the magnetoelastic material. Although magnetic coupling does not define the speed of signal propagation, it should determine the trajectory of spin relaxation. Exceeding the speed of spin wave in ferromagnetic material may lead to a chaotic magnetic reorientation along the ferromagnetic layer. At the same time, it will limit the propagation length. Would it be possible to cascade multiferroic interconnects? This is one of many questions to be answered with further study.

Finally, we wish to compare the main characteristics of different magnetic interconnects and discuss their advantages and shortcomings. Moving a domain wall is a reliable and experimentally proven method for magnetic signal transmission [24]. A domain wall propagates through a magnetic wire, as long as an electric current or an external magnetic field is applied, and remains at a constant position if the driving force is absent. This property is extremely useful for building magnetic memory (e.g., the “racetrack” memory [25]). The speed of domain motion may exceed hundreds of meters per second if the driving electric current has a sufficiently large density (e.g., 250 m/s at 1.5 × 10^8^ A/cm^2^ from [25]). Slow propagation speed and high energy per bit are the main disadvantages of the logic circuits’ exploding domain wall motion.

The interconnects made from the sequence of nano-magnets are relatively faster and less power consuming, where the nearest neighbor nano-magnets are coupled via the dipole–dipole interaction (the so-called Nano-Magnetic Logic (NML) [2]). Experimentally realized wires formed from a line of anti-ferromagnetically ordered nano-magnets show a signal propagation speed up to 10^3^ m/s with an internal (without the losses in the magnetic field generating contours) power dissipation per bit of approximately tens of atto Joules [18]. There is a tradeoff between the speed of signal propagation and the dissipated energy. The slower the speed of propagation, the lower the energy dissipated within the interconnect. The main shortcoming of the nano-magnet interconnect is associated with reliability, as the thermal noise and fabrication-related imperfections can cause errors in signal transmission and the overall logic functionality of the NML circuits [26].

Interconnects exploiting spin waves may provide signal propagation with the speed of 10^4^ m/s–10^5^ m/s. At the same time, the amplitude of the spin-wave signal is limited by the several degrees of magnetization rotation, in contrast to the complete magnetization reversal provided by the domain wall motion or NML. The amplitude of the spin wave decreases during propagation (e.g., the attenuation time for magnetostatic surface spin waves in NiFe is 0.8 ns at room temperature [27]). The unique advantage of the spin-wave approach is that the interconnects themselves can be used as passive logic elements exploiting spin-wave interference. The latter offers an additional degree of freedom for logic gate construction and makes it possible to minimize the number of nano-magnets per logic circuit [4].

The All-Spin Logic (ASL) proposal suggests the use of spin-polarized currents for nano-magnet coupling [3]. This approach allows for much greater defect tolerance, as the variations in the size and position of input/output nano-magnets are of minor importance. It is also scalable, since shorter distances between the input/output cells would require less spin-polarized currents for switching. According to theoretical estimates [28],

ASL can potentially reduce the switching energy-delay product. The major constraint is associated with the need for the spin-coherent channel, where the length of the interconnects exploiting spin-polarized currents is limited by the spin diffusion length.

The described magnetic interconnects based on composite multiferroics combine high transmission speed (as fast as the spin waves) with the possibility of transmitting large amplitude signals (up to 90 degrees of the magnetization rotation). As we stated above, the main appealing property of the proposed interconnect is the ability to keep constant the amplitude of the magnetization signal. All these advantages are the result of using the electro-magnetic coupling in multiferroics, allowing us to pump energy from the electric to the magnetic domain. Based on the presented estimates, the energy per transmitted bit may be as low as several atto Joules per 100 nm of transmitted distance. From a practical point of view, the implementation of composite multiferroic interconnects is feasible, as it relies on the integration of well-known materials (e.g., PMN-PT and Ni) and can be integrated on a silicon platform. However, the dynamics of the electro-mechanical-magnetic coupling in composite multiferroics remain mainly unexplored. The expected challenges are associated with the limited scalability, as the thickness of the piezoelectric should be sufficient to generate the stress required for anisotropy change. The quality of the interface between the piezoelectric and magnetostrictive layers is another important factor to be considered. The inevitable structure imperfections should be below the magnetization reversal threshold (e.g., as defined by Equation (4)). In Table 1, we have summarized the estimates on the main characteristics of different magnetic interconnects and outlined their major advantages and shortcomings.

## 5. Conclusions

In summary, we considered a novel type of magnetic interconnect exploiting electro-magnetic coupling in two-phase composite multiferroics. According to the presented estimates, composite multiferroic interconnects combine the advantages of fast signal propagation (up to 10^5^ m/s) and low power dissipation (less than 1 aJ per 100 nm). The most appealing property of the multiferroic interconnects is the ability to pump energy into the magnetic signal and amplify it during propagation. A voltage-driven magnetic interconnect may be utilized in nano-magnetic logic circuitry and provide an efficient tool for logic gate construction. The fundamental limits and practical constraints inherent to two-phase multiferroics are associated with the efficiency of stress-mediated coupling at high frequencies. There are many questions related to the dynamic of the stress-mediated signal propagation, which will be clarified with a further theoretical and experimental study.

## Figures and Tables

**Figure 1 micromachines-13-01991-f001:**
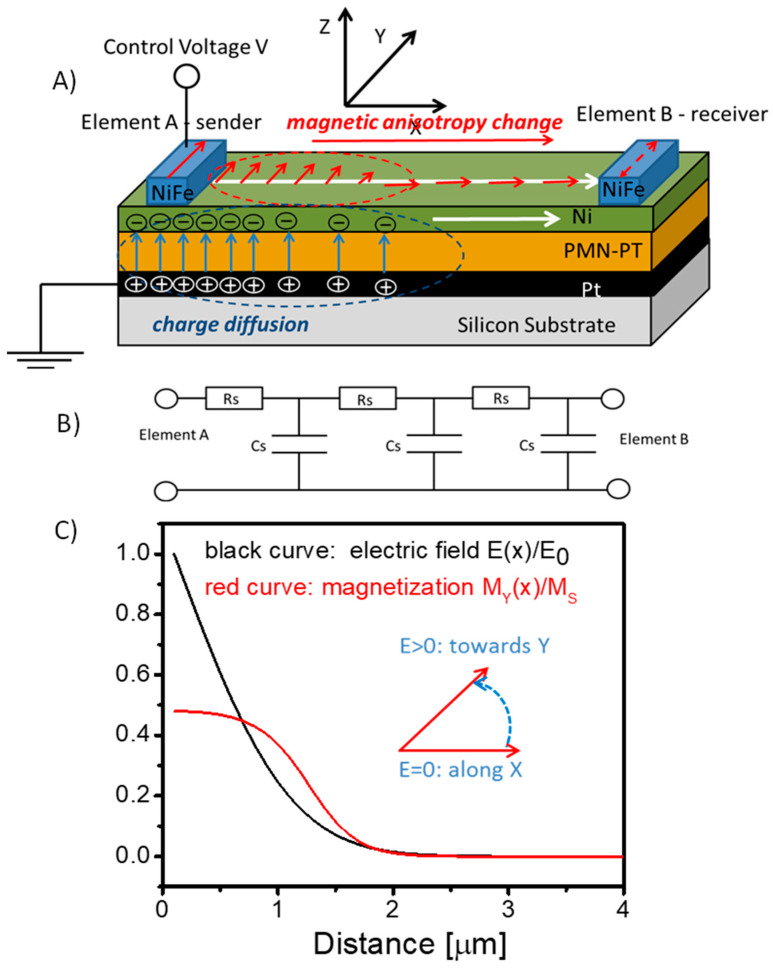
(**A**) Schematics of the synthetic multiferroic interconnect comprising a piezoelectric layer (PMN-PT) and a magnetostrictive layer (Ni). The structure resembles a parallel plate capacitor. An application of voltage at point A results in charge diffusion through the plates. In turn, an electric field applied across the piezoelectric produces stress, which rotates the easy axis of the magnetoelastic material. (**B**) The equivalent electric circuit—RC line, which is used in numerical simulations. (**C**) Results of numerical simulations showing the distribution of the electric field and the magnetization along the interconnect. The change of magnetization in the magnetoelastic layer follows charge diffusion.

**Figure 2 micromachines-13-01991-f002:**
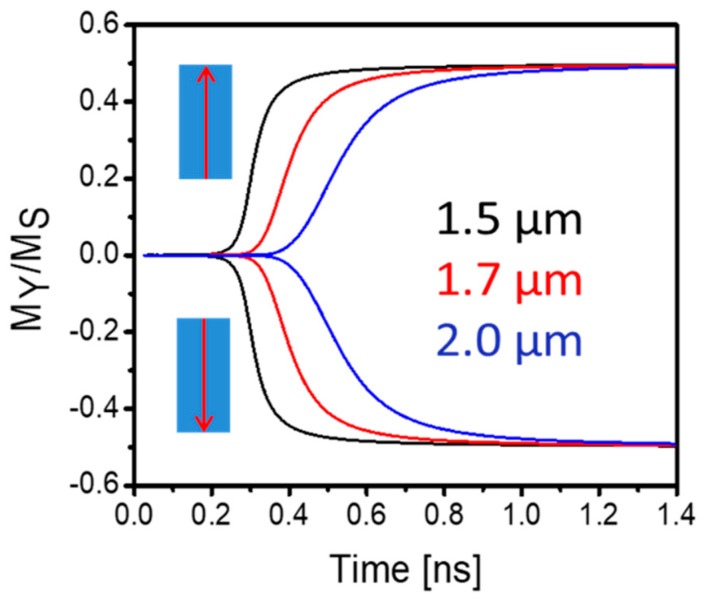
Results of numerical modeling showing the normalized magnetization *M_Y_*/*M_S_* as a function of time. The two sets of curves show magnetization trajectories following the initial state of the sender nano-magnet A (e.g., along or opposite to axis *y*). The black, red, and blue curves show magnetization at 1.0 µm, 2.0 µm, and 3.0 µm distance away from the starting point A.

**Figure 3 micromachines-13-01991-f003:**
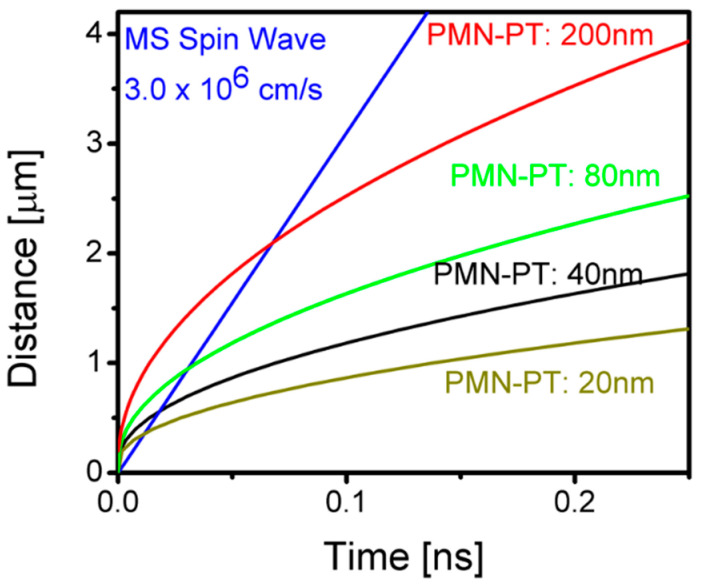
Results of numerical modeling illustrating the speed of signal propagation in the synthetic multiferroic interconnect. Shown are several curves corresponding to different thicknesses of the PMN-PT layer (20 nm, 40 nm, 80 nm, and 200 nm). The blue line is the reference data for the Magnetostatic Surface Spin Wave (MSSW) with a group velocity of 3.0 × 10^4^ m/s.

**Table 1 micromachines-13-01991-t001:** The estimates on the main characteristics of different magnetic.

	Domain Wall	MCA	Spin Wave	ASL	Multiferroics
**Mechanism of coupling**	Domain wall motion	Dipole–dipole coupling	Spin waves	Spin polarized current	Magnetization signal
**Speed of propagation**	10^2^ m/s	10^3^ m/s	10^4^ m/s–10^5^ m/s	* 10^5^ m/s	* 10^5^ m/s
**Energy dissipated per bit transmitted**	>1000 aJ	** 1 aJ	0.1 aJ	N/A	1 aJ
**Main advantage**	Non-volatile, can be stopped at any time and preserve its position	Internal dissipated energy approaches zero at the adiabatic switching	Computation in wires—additional functionality via wave interference	Scalable, defect tolerant	Fast signal propagation, signal amplification
**Main disadvantage**	Slow and energy consuming	Effect of thermal noise increases with the propagation distance	Propagation distance is limited due to the spin-wave damping	Propagation distance is limited by the spin diffusion length	Limited scalability

* Signal propagation speed is determined by the charge diffusion and decreases with the distance. ** The estimates for 10^3^ m/s propagation speed and include only for the energy dissipated inside the magnetic interconnect (without considering the energy losses in the magnetic field generating contours).

## Data Availability

All data generated or analyzed during this study are included in this published article.

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
