# Peer review of "Magnetic Interconnects Based on Composite Multiferroics"

_micromachines, 2022, doi:10.3390/mi13111991_

Round 1

Reviewer 1 Report

The paper "Synthetic Multiferroic Interconnects for Magnetic Logic Circuits" by Alexander Khitun presents the results the possibility of using synthetic multiferroics comprising piezoelectric and magnetostrictive materials as an interconnect for nano-magnetic logic circuits. present the results of numerical modeling illustrating signal propagation through the interconnect. The model combines electric and magnetic parts, where the electric part describes the distribution of an electric field through the piezoelectric and the magnetic part describes the change of magnetization in the magnetoelastic layer.  Synthetic multiferroic interconnects can be implemented in a variety of spin-based devices ensuring reliable and low-energy consuming data transmission. Generally, I think the manuscript can be published in Micromachines after some mandatory improvements which are listed below. I suppose that Author will consider the following recommendations.

1) It would be interesting to include in the article estimates for the magnitude of elastic strain generated in the PMN-PT layer and transmitted to the ferromagnetic layer.

2) If the Author use the idea of synthetic multiferroic structures and mentions interconnections using spin waves, thus the Ref[Journal of Magnetism and Magnetic Materials, 515, 167302 (2020)], Ref[Applied Physics Letters, 118, 26, 262405 (2021)] and [Nanomaterials, 12, 1520 (2022)] should be added.

3) Potentially, what is the maximum length that such synthetic multiferroic interconnections could have? Or could they be as elementary cells? This information may be added through the text.

4) There are stylistic inaccuracies and sloppiness in the article. In particular, in Fig. 2, the designation "2 mm" is written in grey, when logically it should be blue; in lines 271, 281, 330, 337, etc., incorrect values of velocities are indicated (indicated as 103 m/s, etc.).

Reviewer 2 Report

Dear author. The manuscript is interesting and shows a device for magnetic signal transmission. Before publication, it is important to clarify some aspects of its operation.

Text format

T1 - The standard language for multiferroic with two or more phases is composite multiferroic. Why use synthetic multiferroic instead of composite multiferroic?

T2 - Citations are not correctly formatted

T3 - The text does not describe the simulation method and program. Also, it is not specifying device dimensions.

T4 - It is important to put the references of the constants used in the simulation.

T5 - Page 5, line 182. It is not clear who the multiferroic layer is.

T6 - Page 5, line 194. Wrong units

Questions

Q1 - Both electrical polarization and magnetization depend on the direction of the crystal structure. In this case, what are the crystalline directions of PMN-PT and Ni?

Q2 - A point that was not discussed is how the effect of stress between the layers will alter the device's functioning. The used charge diffusion equation does take this into account. Although you consider the approximation made in equation 4 to be a first-order approximation, I believe that stress is an important element. For example, the effect of the magnetostriction of the Ni layer should be considered because of the variation of magnetization along the layer. In this case, the stress between the two materials may change the electric polarization of the ferroelectric layer, changing the charge diffusion. As a consequence, the propagation of the signal will not be that simple and can significantly affect the device's functioning. 

Q3 - Reference 9 discusses the electric field's effect on the Nickel layer's magnetic response. For this, it is considered, for example, the direction of the electric field and where it started (initial state). In your device, are you considering that the initial state is 0 V? How would the cycling be taking into account the polarization of the PMN-PT layer? Does this layer need to be electrically polled?

Q4 – It is necessary to discuss the quality of the interface between the layers since it is important for the device's functioning.

Reviewer 3 Report

Reviewed paper is a theoretical. In general, the paper is informative on contains some interesting results.
Unfortunately, there are many miss, missing references in the tekst done correctly  or  incorrect execution of the reference, e.g. lack of boldone and no indications of new literature reports. The theoretical introduction is insufficient for the reader to delve into the subject of the article.  Another weakness of this work is practically lack of discussion in paragraph and trivial conclusions.

Author Response

We kindly ask the reviewer to take a look at the improved version of the article and the updated reference list. 

Also, it would be important for the reviewer to suggest appropriate and timely references to be included in the text.

Round 2

Reviewer 1 Report

Author significantly improve the manuscript

And now the manuscript could be published 

Reviewer 2 Report

Dear author. Thank you for the clarifications and text changes. The manuscript can be published in the present form.